# N95 respirator hybrid decontamination method using Ultraviolet Germicidal Irradiation (UVGI) coupled with Microwave-Generated Steam (MGS)

Thirumaaran Gopalan[1], Mohd Ridha Muhamad[1,2]*, Victor Chee Wai Hoe[3], Pouya Hassandarvish[4]

1 Department of Mechanical Engineering, Faculty of Engineering, Universiti Malaya, Kuala Lumpur, Malaysia, 2 Centre of Advanced Manufacturing and Material Processing (AMMP Centre), Universiti Malaya, Kuala Lumpur, Malaysia, 3 Centre for Epidemiology and Evidence-Based Practice, Department of Social and Preventive Medicine, Faculty of Medicine, Universiti Malaya, Kuala Lumpur, Malaysia, 4 Tropical Infectious Diseases Research & Education Centre, Universiti Malaya, Kuala Lumpur, Malaysia

* ridha@um.edu.my

**Data Availability Statement:** All relevant data are within the paper and its Supporting Information files.

## Abstract

The Coronavirus Disease 2019 (COVID-19) pandemic has induced a critical supply of personal protective equipment (PPE) especially N95 respirators. Utilizing respirator decontamination procedures to reduce the pathogen load of a contaminated N95 respirator can be a viable solution for reuse purposes. In this study, the efficiency of a novel hybrid respirator decontamination method of ultraviolet germicidal irradiation (UVGI) which utilizes ultraviolet-C (UV-C) rays coupled with microwave-generated steam (MGS) against feline coronavirus (FCoV) was evaluated. The contaminated 3M 1860 respirator pieces were treated with three treatments (UVGI-only, MGS-only, and Hybrid—UVGI + MGS) with variable time. The virucidal activity was evaluated using the $TCID_{50}$ method. The comparison of decontamination efficiency of the treatments indicated that the hybrid method achieved at least a pathogen log reduction of 4 logs, faster than MGS and UVGI. These data recommend that the proposed hybrid decontamination system is more effective comparatively in achieving pathogen log reduction of 4 logs.

## Introduction

The unprecedented Coronavirus Disease 2019 (COVID-19) pandemic has alarmed the world regarding the need for respiratory protection against airborne diseases. This urgent necessity for respiratory protection resulted in a rapid increase in demand for personal protective equipment (PPE), particularly face masks. Healthcare professionals who often handle COVID-related illnesses rely solely on PPE, particularly N95 respirators, to safeguard against this virus [1]. The N95-type respirator is applicable and widely used in healthcare applications [2].

Consequently, a respirator decontamination technique for reuse could assist in resolving the rising demand for respirator supply. In order to increase the reusability of respirators, a

**Funding:** This research was funded by the Universiti Malaya COVID-19 Related Special Research Grant (UMCSRG - CSRG003-2020ST). The funders had no role in study design, data collection, analysis, manuscript preparation, or the decision to publish.

**Competing interests:** The authors have declared that no competing interests exist.

safe respirator decontamination technique may help reduce the pathogen burden of a contaminated respirator. The 3M company indicates that reducing the pathogen burden of the respirator through a decontamination procedure is one of the key criteria for reusing a respirator [3]. According to the National Institute of Occupational Safety and Health (NIOSH), UVGI, hydrogen peroxide vapor (HPV), and moist heat are recommended to be efficient decontamination treatments against pathogens [4].

Since the late 19th century, UVGI treatment has been used and practiced in disinfection applications all over the world [5]. UVGI procedure utilizes a UVC-producing source such as bulbs to kill viral, bacterial, and fungal organisms [6]. By causing damage to their DNA, UV-C radiation either kills or inactivates microorganisms [5]. The degradation of DNA structure by UV-C can be due to the formation of photoproducts, free radicals generation, and strand breaks which include either single or double DNA strand breaks [7, 8]. Related investigations demonstrate the UVGI decontamination system's effectiveness against coronaviruses like SARS-CoV-2 and MERS-CoV [9, 10]. Ozog et al. produced excellent virus inactivation results of at least 3 log reductions in viable SARS-CoV-2 using a dosage of 1.5 J/cm$^2$ [11]. Furthermore, a log reduction of at least 3 was achieved by Geldert et al. where the specimen was treated by 0.05–0.5 J/cm$^2$ of UVC dosage [12]. High virucidal activity while maintaining the integrity of the treated specimen is one of the key advantages of the UVGI treatment. The degradation of N95 respirators associated with UVGI treatment can be explained by the administration of excessive UV dosage to a specimen. Several reports exhibit excellent log reduction results produced while preserving the integrity and performance of a treated mask [11, 13–17]. In addition, an excessive UVC dosage of 950 J/cm$^2$ on the 3M 1860 N95 respirator was tested by Lindsley et al. This study concludes that at the specified high dosage the filtration performance was minimally affected and flow resistance was not affected [14].

MGS is among the decontamination techniques investigated currently that show great potential. MGS procedure is a type of steam treatment that uses microwave oven sterilization coupled with the presence of moisture. Water is capable of absorbing microwave energy, lowering the risk of harming N95 respirator components [18]. Notably, MGS treatment is well known for quick disinfection and affordability where microwave oven has been a common household item nowadays. In this treatment, pathogens are inactivated by generating heat within water molecules using microwave energy. Initial studies demonstrate that MGS was capable of producing outstanding microbiological outcomes with the preservation of the integrity of the treated respirators at a relatively shorter treatment time [19–22]. Fisher et al. reported an excellent 3-log reduction in MS2 using a 90-second microwave treatment. This study described an MGS procedure using steam bags and a commercially available microwave oven. Notably, filtration performance was preserved post-decontamination [19].

The purpose of this study was to evaluate the decontamination efficiency of the proposed hybrid decontamination system (UVGI coupled with MGS) in treating an N95 respirator which contaminated with feline coronavirus (FCoV). Contaminated pieces of N95 respirator were subjected to 3 different decontamination conditions (UVGI-only, MGS-only, and Hybrid—UVGI + MGS) which were analyzed through a virus viability test.

## Materials and methods

### Design of the study

This specific study is an experimental research. An extensive literature review was conducted on various respirator decontamination techniques which include UVGI, MGS, HPV, dry heat, moist heat, and ethanol [1]. The selection of UVGI and MGS was based on the ease of incorporating both treatments in one cycle and excellent decontamination efficiency against

pathogens. The parameter design was developed using estimation from existing reported data in the reviewed articles.

## Respirator

The N95 respirator model selected for this experiment was 3M™ 1860 (3M™ Health Care Particulate Respirator and Surgical Mask 1860, 3M, USA). This specific model was selected based on its wide application in healthcare settings specifically by health workers. An N95 respirator has four layers: the coverweb, the shell, the first filter, and the second filter. Polypropylene is used to make the filter layers, whereas polyester is used to make the coverweb and shell layers [2]. Rectangular pieces of measurement are approximately 4 cm x 2 cm (LxH) which accounts for 4.8% of the whole respirator excised from the respirators.

## Cells and virus

In this study, feline coronavirus (FCoV) obtained from the Tropical Infectious Diseases Research & Education Centre was used. The FCoV was propagated in the CRFK cell line using Dulbecco's Modified Eagle Medium (DMEM) (HyClone, Thermo Scientific, USA) supplemented with 2% Fetal Bovine Serum (FBS) (Gibco, Thermo Scientific, USA). Following the observation of cytopathic effects in the cells, the virus was harvested and stored at -80˚C for future use. The viral titers were determined using the $TCID_{50}$ method.

## Viral kill time assay

To perform the viral inactivation assay, each respirator piece (4x2 cm) was covered with 100 μl of FCoV suspension with a titre of $1x10^6$ TCID50/ML according to the test standard ISO18184-2019 [23]. Subsequently, the infected respirators were subjected to three different treatments for disinfection. These treatments included exposure to UVGI, MGS, or a hybrid procedure using a combination of UVGI and MGS at different time points. A control group, which was not exposed to any treatments, was included for comparison purposes.

To recover the virus from the infected respirators, rinsing was performed using 1 ml of DMEM containing 2% FBS. The recovered virus was then transferred to a microcentrifuge tube, and a 10x serial dilution was performed. The different dilutions of the virus were used to infect the CRFK cells seeded in a 96-well plate. The cells were incubated for 72 hours until cytopathic effects (CPE) developed. Fixation and staining of the infected cells were carried out using a mixture of paraformaldehyde and crystal violet. The virus titer was determined using the Spearman-Karber method and expressed as a tissue culture infectious dose 50% ($TCID_{50}$/ml). The virucidal activity was assessed by calculating the difference in logarithmic titer ($\Delta \log_{10} TCID_{50}$/ml) between the virus control and the test virus.

## Decontamination treatments

A total of 3 types of decontamination treatments were investigated in this research which include UVGI-only, MGS-only, and Hybrid (UVGI + MGS). The detailed operational procedures of these treatments are available in the S1 File.

## UVGI decontamination system

A custom mask sterilizer machine equipped with a UVGI decontamination system measuring 270 x 315 x 470 mm (LxHxW) was designed and fabricated for this study (Fig 1A). The machine was made up of Stainless Steel 304 for the majority of the fabricated parts where a few small parts were fabricated using Aluminium 5083. A special mask enclosure was made using

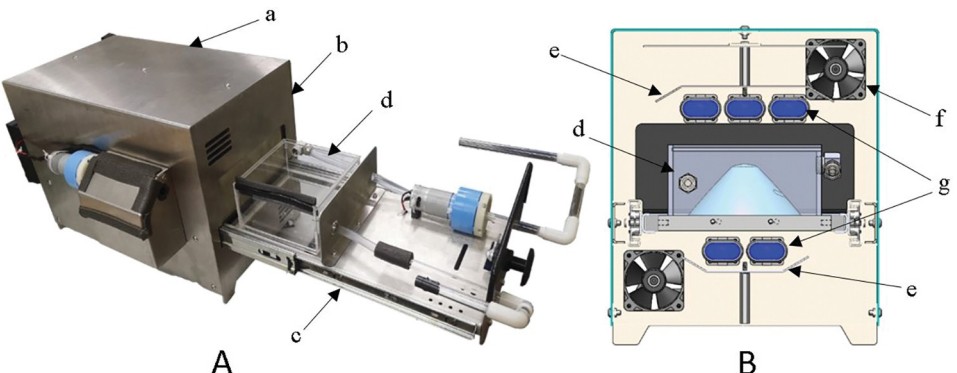

**Fig 1. Mask sterilizer machine.** (A) Working prototype: a, cover; b, main body; c, drawer; d, quartz mask case; (B) 3D Model: e, reflector; f, dc fan; g, UVC bulbs.

acryl and quartz glass. The selection of quartz glass was based on its ability to allow UVC rays to penetrate through and reach the contaminated specimen. The design of the UVGI system in the machine was constructed using 5 UVC bulbs (TUV PL-S 9W; Phillips)), capable of producing rays with a peak at 253.7 nm. These bulbs were arranged in a manner where 3 focus on the exterior meanwhile 2 focus on the interior of the respirator specimen (Fig 1B). Only 3 upper bulbs were utilized for this study which generates a combined irradiance value of at least 7.5 mW/cm$^2$ at 10 cm. Polished reflective stainless steel sheets were placed around the bulbs to maximize the reflection of ultraviolet rays. Eq 1 was used to calculate the required UV dosage for each designed parameter of the tests [24].

$$UV\ Dose\left(\frac{J}{cm^2}\right) = Irradiance\left(\frac{W}{cm^2}\right) x\ Time(s) \tag{Eq1}$$

## MGS decontamination system

A commercially available microwave oven (MG23K3513GK; Samsung) was selected for MGS-based decontamination tests. This 1200W-rated microwave oven is capable of producing an output power of 600–650 W (S2 File). This microwave is equipped with a turntable which ensures uniform heating and steam generation [22]. The decontamination setup (Fig 2) for

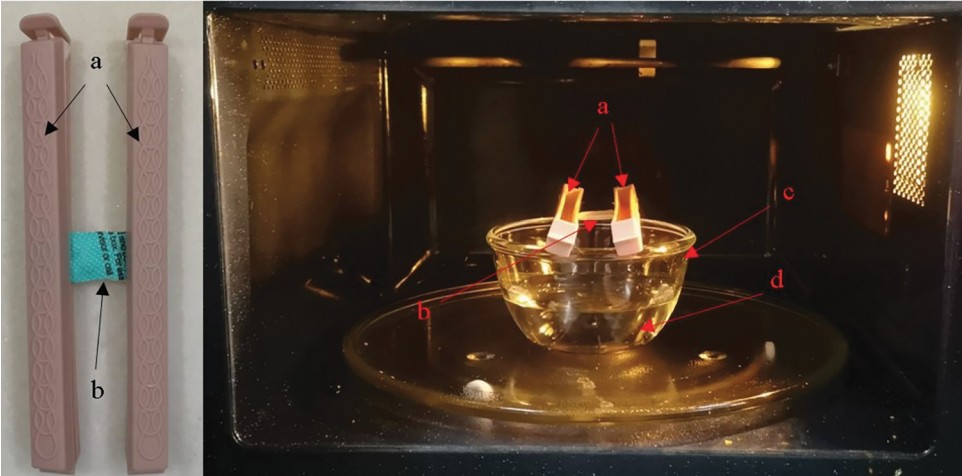

**Fig 2. Microwave-Generated Steam (MGS) treatment setup.** a, sealer clips; b, excised N95; c, glass bowl; d, water.

**Table 1. Virus viability test results (UVGI-only).**

| UVGI-only | | | |
|---|---|---|---|
| Sample | Exposure Time (s) | Dosage (J/cm$^2$) | Log Reduction (log$_{10}$) |
| UV Control | 0 | 0 | 0 |
| UV 1 | 3 | 0.0225 | 1 |
| UV 2 | 6 | 0.045 | 4 (Omitted) |
| UV 3 | 9 | 0.0675 | 3 |
| UV 4 | 17 | 0.1275 | 3 |
| UV 5 | 34 | 0.255 | 2 (Omitted) |
| UV 6 | 67 | 0.5025 | 3 |
| UV 7 | 100 | 0.75 | $\geq 4$ |

this technique was inspired by the investigation by Zulauf et al. [22]. The inoculated respirator piece was fixed in between 2 parallel sealer clips. This fixture was suspended over a bowl of 150 ml water which was subjected to a specified microwave treatment time.

## Results

The maximum detectable log reduction for this test was determined to be a 4 log reduction and each sample was tested three times to produce one log reduction data for the specific sample. The UVGI-only tests resulted in log reduction exceeding the detection limit of 4 logs after being treated with a UVC dosage of 0.75 J/cm$^2$ (Table 1). This UVC dosage is equivalent to 100 seconds of UV exposure. In addition, 3 log reduction was observed at a dosage of 0.0675 J/cm$^2$. 2 data were omitted due to ambiguity.

A total of 5 time points were tested for MGS-only treatment. Inoculated N95 respirator exposed to 45 seconds of MGS treatment exhibits a log reduction of 4 (Table 2). Moreover, 2 log reduction was achieved using 30 seconds of exposure.

Next, we evaluated the hybrid decontamination tests of UVGI coupled with MGS. MGS exposure was fixed at 30 s meanwhile UV exposure varied in the range of 3 to 34s. At a total exposure time of 39 s, a log reduction of at least 4 was achieved (Table 3). 36 s of hybrid decontamination system exposure exhibit a 3 log reduction in pathogen load.

## Discussion

In this study, the efficiency of a hybrid respirator decontamination technique of UVC coupled with MGS against FCoV was evaluated in comparison to single treatments of UVGI and MGS. Results from the tests indicate excellent virucidal activity incorporated with the proposed hybrid decontamination technique. The hybrid decontamination technique achieved pathogen reduction of at least 4 logs faster than the single treatments of UVGI and MGS (Fig 3). The

**Table 2. Virus viability test results (MGS-only).**

| MGS-only | | |
|---|---|---|
| Sample | Exposure Time (s) | Log Reduction (log$_{10}$) |
| MGS Control | 0 | 0 |
| MGS 1 | 15 | 0 |
| MGS 2 | 30 | 2 |
| MGS 3 | 45 | $\geq 4$ |
| MGS 4 | 60 | $\geq 4$ |

**Table 3. Virus viability test results (Hybrid—UVGI + MGS).**

| Sample | Hybrid (UVGI + MGS) | | | | |
|---|---|---|---|---|---|
| | UVGI | | MGS | Total Exposure Time (s) | Log Reduction (log10) |
| | Exposure Time (s) | Dosage (J/cm$^2$) | Exposure Time (s) | | |
| Control | 0 | 0 | 0 | 0 | 0 |
| HYB 1 | 3 | 0.0225 | 30 | 33 | 2 |
| HYB 2 | 6 | 0.045 | 30 | 36 | 3 |
| HYB 3 | 9 | 0.0675 | 30 | 39 | $\geq 4$ |
| HYB 4 | 17 | 0.1275 | 30 | 47 | $\geq 4$ |
| HYB 5 | 34 | 0.255 | 30 | 64 | $\geq 4$ |

reported $\geq 4$ log reduction in this current study is considered significant as 3 logs are the minimum required pathogen reduction to completely disinfect a contaminated filtering facepiece respirator (FFR) [24, 25].

The UVGI-only test in this research produced significant microbial load reduction in FCoV of at least 3 log reduction at a UVC dosage of 0.0675 J/cm$^2$. This result is consistent with the study reported by Geldert et al. where a log reduction of 3 was achieved using 0.05–0.5 J/cm$^2$ of UVC dosage in viable SARS-CoV-2 on a 3M 1860 coupon [12]. Multiple investigations further validate the 4 log reduction achieved in this study where higher log reduction was achieved using higher UVC dosages [12, 16, 26]. In addition, MGS-only treatment reported excellent virucidal results where a 4 log reduction was achieved at 45 seconds of exposure which is in agreement with Zulauf et al. [22]. Comparing single treatments utilized in this study, UVGI achieves 3 log reduction faster than MGS, while MGS achieves 4 log reduction significantly faster than UVGI.

To our knowledge, this current study is the first to report an evaluation of the efficiency of the hybrid decontamination (UVGI + MGS) on an N95 respirator against FCoV. The key idea

**Fig 3. Graph of log reduction value against treatment time (Single vs Hybrid).**

of approaching this hybrid decontamination technique in this study was to find a treatment that offers rapid and safe decontamination. As the results suggest, hybrid decontamination using UVGI and MGS can significantly save time and cost in decontaminating respirators up to a safe level. Both UVGI and MGS exhibit great potential as efficient decontamination systems where a higher log reduction value is offered at a relatively shorter treatment time. In comparison with the other available treatments, these two system does not leave any residue which could pose a safety threat to the user. However, several disadvantages of these systems have to be handled with caution. Firstly, as the previous studies suggest, each decontamination procedure has its drawback of affecting the integrity and performance of the respirator when exposed to longer treatment times [14, 20, 22, 27]. Therefore, multiple decontamination cycles must be monitored closely. The hybrid treatment design could potentially alleviate the dependence on a single system for complete sterilization which could assist in preserving the functionality of the respirator for reuse. Secondly, a basic level of expertise is required in handling the UVGI device. In addition, the limited space of the microwave oven constricts the number of respirators that can be decontaminated in one cycle [1].

The considerations for the future scope of this current study need to be noted. Firstly, the current study does not report on performance and integrity degradation incorporated with respirator decontamination. Referring to UVGI decontamination on 3M 1860 respirator, Lindsley et al. reported unaffected flow resistance at UV dosages up to 950 J/cm$^2$ [14], and Lore et al. reported no signification changes to filter performance at a dosage of 1.8 J/cm$^2$ [20]. Referring to MGS decontamination on 3M 1860 respirator, Fisher et al. reported preserved filtration efficiency after 1 decontamination cycle of 90 seconds [19], and Zulauf et al. reported preservation of fit, seal, and filtration performance for up to 20 cycles of 180 seconds [22]. In comparison, our current study utilizes significantly lower UV dosages and MGS treatment time which could eliminate the possibility of significant integrity or performance impact on the respirator. Secondly, the irradiance value of a UVC source which indicates the energy per unit area directly proportional to the UV dosage value. Sufficient dosage value is crucial for complete decontamination. Future investigations could utilize UVC setup with higher irradiance values which could offer relatively shorter treatment time to achieve specific log reduction values. Thirdly, the resistance of the tested organism plays a major role in the dosage of treatment required for complete decontamination. FCoV used in this study which is an enveloped virus is one of the least resistant organisms against disinfectants. More resilient microorganisms such as non-enveloped viruses and bacterial spores may require higher treatment time for complete decontamination [28].

There are limitations to be addressed in this current study. Firstly, decontamination treatment was performed on the excised pieces of the N95 respirator rather than the whole N95 respirator. Nevertheless, we expect the results will same or even better when the whole respirator is decontaminated. Second, only the exterior part of the respirator was tested in this study. In order for the system to be implemented in healthcare applications, further investigation on the interior surface will be needed.

## Conclusion

Effective FFR decontamination procedures are crucial in critical pandemic scenarios such as COVID-19. UVGI and MGS exhibit great potential in treating contaminated N95 respirators. Our study concludes that the hybrid decontamination procedure of UVGI coupled with MGS is more effective than single treatments of UVGI and MGS in achieving a 4-log pathogen load reduction. In addition, UVGI was able to produce a 3-log reduction in a quarter of the time managed by the hybrid decontamination procedure. Implementation of this procedure will

require further investigations on models, filtration performance, and fit testing of the treated respirators.

## Supporting information

**S1 File. Decontamination system operation procedure.**
(DOCX)

**S2 File. Determination of microwave power output.**
(XLSX)

## Acknowledgments

The authors would like to thank Mohd Fauzi Bakri Hashim and Tropical Infectious Diseases Research & Education Centre (TIDREC) for their assistance in this research project.

## Author Contributions

**Conceptualization:** Thirumaaran Gopalan, Mohd Ridha Muhamad, Victor Chee Wai Hoe, Pouya Hassandarvish.

**Data curation:** Thirumaaran Gopalan.

**Formal analysis:** Thirumaaran Gopalan.

**Investigation:** Thirumaaran Gopalan.

**Methodology:** Thirumaaran Gopalan.

**Supervision:** Mohd Ridha Muhamad, Victor Chee Wai Hoe.

**Validation:** Mohd Ridha Muhamad, Victor Chee Wai Hoe.

**Writing – original draft:** Thirumaaran Gopalan, Pouya Hassandarvish.

**Writing – review & editing:** Thirumaaran Gopalan, Mohd Ridha Muhamad.

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
