## [Decision Letter · Decision Letter 0]

9 Nov 2023

PONE-D-23-26467N95 Respirator Hybrid Decontamination Method Using Ultraviolet Germicidal Irradiation (UVGI) Coupled with Microwave-Generated Steam (MGS)PLOS ONE

Dear Dr. Gopalan,

Thank you for submitting your manuscript to PLOS ONE. After careful consideration, we feel that it has merit but does not fully meet PLOS ONE’s publication criteria as it currently stands. Therefore, we invite you to submit a revised version of the manuscript that addresses the points raised during the review process.

In detail, your paper has been carefully evaluated  by two experts in the filed and I fully agree with their suggestions and modification proposals.

We look forward to receiving your revised manuscript.

Kind regards,

Vittorio Sambri, M.D., Ph.D.

Academic Editor

PLOS ONE

Journal Requirements:

Reviewers' comments:

Reviewer's Responses to Questions

**Comments to the Author**

1. Is the manuscript technically sound, and do the data support the conclusions?

Reviewer #1: Yes

Reviewer #2: Yes

2. Has the statistical analysis been performed appropriately and rigorously? 

Reviewer #1: I Don't Know

Reviewer #2: I Don't Know

3. Have the authors made all data underlying the findings in their manuscript fully available?

Reviewer #1: No

Reviewer #2: Yes

4. Is the manuscript presented in an intelligible fashion and written in standard English?

Reviewer #1: Yes

Reviewer #2: Yes

5. Review Comments to the Author

Reviewer #1: Thank you for the opportunity to review this interesting paper on decontamination of N95 respirators. Below are my recommendations for areas that should be addressed before publication.

When introducing the term UVGI in the abstract, specify the type of UV eg UVC

The use of the word ‘mask’ is confusing. If referring to an N95 respirator, the term ‘respirator’ would be more appropriate and accurate.

Remember to define acronyms eg. COVID

On Pg.4 – the term pathogens are denatured is used. Feels. Proteins are denatured and pathogens are typically described as inactivated, killed, lysed, etc.

Interesting mention of the drawbacks on page 4. I would recommend moving these to the discussion.

Viral Kill Time Assay methods – please describe how the masks were ‘infected’. Eg 100 ul was pipetted over the entire surface of the mask piece?

There needs to be a description in the methods of how the dual exposure to MGS and UVGI happened. Was it sequential? Which treatment occurred first?

How is the hybrid method is a significant time or cost savings as purported by the authors? Hybrid method: 3 logs was 36 seconds, 4 logs 39 seconds. MGS: 4 logs was 45 seconds; UVGI 3 logs after 9 seconds and 4 logs after 100 seconds. Without a thorough explanation of the hybrid method procedure, it is hard to imagine the total operating time would be less using the hybrid method, including operator time to operate two different decon machines. Please provide an explanation.

Reviewer #2: Title

- The title of the manuscript is suitable and clear.

Abstract

- Abstract is an adequate summary of the work presented.

Introduction

- The word sterilization should be changed in disinfection when you use UVC

- “3M indicates that reducing”: I think it is useful to write 3M company

- “HPV” : an acronym should be written out in full the first time it is used.

- “By causing damage to their DNA, UV-C radiation either kills or inactivates microorganisms (5).” : On this subject, I believe there are more effective citations, for example Kciuk M, Marciniak B, Mojzych M, Kontek R. Focus on UV-Induced DNA Damage and Repair-Disease Relevance and Protective Strategies. Int J Mol Sci. 2020 Oct 1;21(19):7264. doi: 10.3390/ijms21197264. PMID: 33019598; PMCID: PMC7582305 or Rastogi RP, Richa, Kumar A, Tyagi MB, Sinha RP. Molecular mechanisms of ultraviolet radiation-induced DNA damage and repair. J Nucleic Acids. 2010 Dec 16;2010:592980. doi: 10.4061/2010/592980. PMID: 21209706; PMCID: PMC3010660.

- “In addition, the limited space of the microwave oven constricts the number of masks that can be decontaminated in one cycle (1).” : Is the number in brackets a citation or is it the number of masks that can be decontaminated in one cycle?

Method

- The timing of the study (start, run and end) should be described.

- The design/type of the study is missing

- “5 UVC bulbs (TUV PL-S 9W; Phillips)” : I think it is useful to specify the wavelength. The manufacturer states a peak of 253.7 nm in the data sheet, does the value in the prototype correspond to this?

- “This 1200W-rated microwave oven is capable of producing an output power of 600 - 650 W.” : It is useful to specify the power used for the experiment, as the product data sheet states a maximum output power of 800 Watts with 6 different levels.

- There is no description of how the three tests are carried out, particularly in relation to the combined test. Which decontamination method is used first in the combined test? Was there any downtime between methods? In my opinion, a step-by-step description of the procedure for the three tests is necessary.

- Repetitions of the tests are not present

- Provide a more detailed description of the decontamination techniques used. Adding explanatory figures about the techniques employed and the execution protocols could make the reading more understandable for the readers.

- Include the brand names of the culture media and solutions used, along with the legal headquarters of the manufacturing company.

Results

- Table 1 is not consistent with the log reduction. The results are inaccurate. How many times were repeated the tests?

Discussion

- “FFR” : An acronym should be written out in full the first time it is used.

- it should be verified what would happen, using microwaves, to the masks on a mechanical level. I believe that heating could have a negative impact. It is plausible that there could be an alteration in the fiber structure. This aspect should be investigated.

- The analyzed sample is relatively small. Is it conceivable that the distribution of microwaves inside the chamber is uniform? If not, there could be a different level of disinfection on the mask.

- Another aspect that should be investigated is the energy distribution of UVC light inside the chamber when a mask is inserted. Would the energies at play per unit of surface area change? These aspects should be commented on.

- The research is solely based on testing one virus. The resistance to other pathogens could be at least estimated based on the literature and the presented results. The hypothesis of the technique used is that it may be effective with other microbial species as well.

- With regard to the limitations of the work, more data would have given the study greater significance and, in addition, repeating the tests on several samples for each exposure time would have avoided the ambiguity of the data found in the UVGI-only decontamination test.

Conclusion

- It should be noted: UVGI reaches 3 logs (minimum required pathogen reduction to

completely disinfect a contaminated) in about a quarter of the time compared to UVGI + MGS.

References

- Cited References are relevant and sufficient to place the work in context. (Have a look at the suggested work in the Introduction section.)

Tables and Figures

- The tables and graph are clear. Graph needs better quality

- For the sake of clarity, the legend should be reproduced below the relevant figure.

6. PLOS authors have the option to publish the peer review history of their article (what does this mean?). If published, this will include your full peer review and any attached files.

Reviewer #1: **Yes: **Jesse Cooper

Reviewer #2: **Yes: **Gabriele Messina

---

## [Author Response · Author response to Decision Letter 0]

9 Dec 2023

Thank you for giving us the opportunity to re-submit our manuscript. The manuscript has been revised according to the reviewers’ comments, as follows:

Reviewer #1: 

Question 1: When introducing the term UVGI in the abstract, specify the type of UV eg UVC.

Answer: Dear reviewer, thank you for your comment. The abstract (Pg 2) section has been modified according to the suggestion.

Question 2: The use of the word ‘mask’ is confusing. If referring to an N95 respirator, the term ‘respirator’ would be more appropriate and accurate.

Answer: Dear reviewer, thank you for your comment. The suggestion is apt and accurate. The entire revised manuscript has been modified in line with the term respirator.

Question 3: Remember to define acronyms eg. COVID

Answer: Dear reviewer, thank you for your comment. The first instance of the COVID term has been defined in the abstract (Pg 2) and introduction (Pg 2) section.

Question 4: On Pg.4 – the term pathogens are denatured is used. Feels. Proteins are denatured and pathogens are typically described as inactivated, killed, lysed, etc.

Answer: Dear reviewer, thank you for your comment. The suggestion is apt and accurate. The introduction (Pg 4) section has been modified according to the suggestion.

Question 5: Interesting mention of the drawbacks on page 4. I would recommend moving these to the discussion.

Answer: Dear reviewer, thank you for your comment.The details on the drawbacks of the decontamination systems have been moved to the discussion (Pg 10-11) section.

Question 6: Viral Kill Time Assay methods – please describe how the masks were ‘infected’. Eg 100 ul was pipetted over the entire surface of the mask piece?

Answer: Dear reviewer, thank you for your comment. The materials and methods (Pg 6) section has been modified according to the suggestion.

Question 7: There needs to be a description in the methods of how the dual exposure to MGS and UVGI happened. Was it sequential? Which treatment occurred first?

Answer: Dear reviewer, thank you for your comment. Since this comment is similar to Question 14 from Reviewer 2, the answer is consolidated together and discussed more in detail. Kindly please review the comment in the said section.

Question 8: How is the hybrid method is a significant time or cost savings as purported by the authors? Hybrid method: 3 logs was 36 seconds, 4 logs 39 seconds. MGS: 4 logs was 45 seconds; UVGI 3 logs after 9 seconds and 4 logs after 100 seconds. Without a thorough explanation of the hybrid method procedure, it is hard to imagine the total operating time would be less using the hybrid method, including operator time to operate two different decon machines. Please provide an explanation.

Answer: Dear reviewer, thank you for your comment. A detailed flow chart of the operational procedure has been added in the supporting information section denoted as S1 File. Decontamination system operation procedure. The authors would like to clarify that this study is a fundamental study which scope is to find the least decontamination time possible using the hybrid method in comparison with other single treatments. In that case, the hybrid method offers faster treatment time in comparison. The reported 6-second advantage (hybrid compared to MGS 4-log) could be significantly higher in the real application of multiple cycles or a higher number of respirators decontaminated in comparison with other methods. In addition, significantly shorter treatment times could result in lower overall operation costs. Nevertheless, this real application was not included in the scope of the current research.

Reviewer #2: 

Question 1: The title of the manuscript is suitable and clear. (Title)

Answer: Dear reviewer, thank you for your comment.

Question 2: Abstract is an adequate summary of the work presented. (Abstract)

Answer: Dear reviewer, thank you for your comment.

Question 3: The word sterilization should be changed in disinfection when you use UVC (Introduction)

Answer: Dear reviewer, thank you for your comment. The suggestion is apt and accurate. The introduction (Pg 3-4) section has been modified according to the suggestion.

Question 4: “3M indicates that reducing”: I think it is useful to write 3M company (Introduction)

Answer: Dear reviewer, thank you for your comment. The suggestion is apt and accurate. The correct term has been modified.

Question 5: “HPV” : an acronym should be written out in full the first time it is used. (Introduction)

Answer: Dear reviewer, thank you for your comment. The first instance of the HPV term has been defined in the introduction (Pg 3) section.

Question 6: “By causing damage to their DNA, UV-C radiation either kills or inactivates microorganisms (5).” : On this subject, I believe there are more effective citations, for example Kciuk M, Marciniak B, Mojzych M, Kontek R. Focus on UV-Induced DNA Damage and Repair-Disease Relevance and Protective Strategies. Int J Mol Sci. 2020 Oct 1;21(19):7264. doi: 10.3390/ijms21197264. PMID: 33019598; PMCID: PMC7582305 or Rastogi RP, Richa, Kumar A, Tyagi MB, Sinha RP. Molecular mechanisms of ultraviolet radiation-induced DNA damage and repair. J Nucleic Acids. 2010 Dec 16;2010:592980. doi: 10.4061/2010/592980. PMID: 21209706; PMCID: PMC3010660. (Introduction)

Answer: Dear reviewer, thank you for your comment. Excellent suggestion of articles. The new articles have been added to the introduction (Pg 3-4) section.

Question 7: “In addition, the limited space of the microwave oven constricts the number of masks that can be decontaminated in one cycle (1).” : Is the number in brackets a citation or is it the number of masks that can be decontaminated in one cycle? (Introduction)

Answer: Dear reviewer, thank you for your comment. The number in the bracket refers to a citation (Gopalan T, Mohd Yatim RaA, Muhamad MR, Mohamed Nazari NS, Awanis Hashim N, John J, et al. Decontamination Methods of N95 Respirators Contaminated with SARS-CoV-2. Sustainability. 2021;13(22):12474). This statement has been moved to the discussion (Pg 10-11) section as per the suggestion by Reviewer 1.

Question 8: The timing of the study (start, run and end) should be described. (Method)

Answer: Dear reviewer, thank you for your comment. Since this comment is similar to Question 14 from Reviewer 2, the answer is consolidated together and discussed more in detail. Kindly please review the comment in the said section.

Question 9: The design/type of the study is missing (Method)

Answer: Dear reviewer, thank you for your comment. The design of the study has been added to the materials and methods (Pg 5) section.

Question 10: “5 UVC bulbs (TUV PL-S 9W; Phillips)” : I think it is useful to specify the wavelength. The manufacturer states a peak of 253.7 nm in the data sheet, does the value in the prototype correspond to this? (Method)

Answer: Dear reviewer, thank you for your comment. A measurement process of the wavelength of the UV rays was not performed. This is due to the bulbs utilized in this prototype were newly acquired. Therefore, all the specifications and calculations were based on the manufacturer’s data sheet which indicates a peak of 253.7 nm. The additional details regarding the wavelength of the bulbs have been modified in the materials and methods (Pg 7) section.

Question 11: “This 1200W-rated microwave oven is capable of producing an output power of 600 - 650 W.” : It is useful to specify the power used for the experiment, as the product data sheet states a maximum output power of 800 Watts with 6 different levels. (Method)

Answer: Thank you for the comment. The output power of 600 - 650 W as stated in the manuscript was calculated using the determination of microwave power output method according to IEC 60705. The formula and data have been attached in the supporting information section denoted as S2 File. Determination of microwave power output.

Question 12: There is no description of how the three tests are carried out, particularly in relation to the combined test. Which decontamination method is used first in the combined test? Was there any downtime between methods? In my opinion, a step-by-step description of the procedure for the three tests is necessary. (Method)

Answer: Dear reviewer, thank you for your comment. Since this comment is similar to Question 14 from Reviewer 2, the answer is consolidated together and discussed more in detail. Kindly please review the comment in the said section.

Question 13: Repetitions of the tests are not present (Method)

Answer: Dear reviewer, thank you for your comment. Since this comment is similar to Question 16 from Reviewer 2, the answer is consolidated together and discussed more in detail. Kindly please review the comment in the said section.

Question 14: Provide a more detailed description of the decontamination techniques used. Adding explanatory figures about the techniques employed and the execution protocols could make the reading more understandable for the readers. (Method)

Answer: Dear reviewer, thank you for your comment. A detailed flow chart of the operational procedure has been added in the supporting information section denoted as S1 File. Decontamination system operation procedure.

Question 15: Include the brand names of the culture media and solutions used, along with the legal headquarters of the manufacturing company. (Method)

Answer: Dear reviewer, thank you for your comment. The materials and methods (Pg 5-6) section has been modified according to the suggestion.

Question 16: Table 1 is not consistent with the log reduction. The results are inaccurate. How many times were repeated the tests? (Results)

Answer: Dear reviewer, thank you for your comment. We would like to confirm that each sample was tested 3 times and we have a few repeated data for one specific log reduction data. We did not repeat all the tests as it is expensive. However, data from hybrid test results confirmed repeatability.

Question 17: “FFR” : An acronym should be written out in full the first time it is used. (Discussion)

Answer: Dear reviewer, thank you for your comment. The first instance of the FFR term has been defined in the discussion (Pg 9-10) section.

Question 18: it should be verified what would happen, using microwaves, to the masks on a mechanical level. I believe that heating could have a negative impact. It is plausible that there could be an alteration in the fiber structure. This aspect should be investigated. (Discussion)

Answer: Dear reviewer, thank you for your comment. The discussion (Pg 11-12) section has been modified according to the suggestion.

Question 19: The analyzed sample is relatively small. Is it conceivable that the distribution of microwaves inside the chamber is uniform? If not, there could be a different level of disinfection on the mask. (Discussion)

Answer: Dear reviewer, thank you for your comment. The materials and methods (Pg 7-8) section has been modified according to the suggestion.

Question 20: Another aspect that should be investigated is the energy distribution of UVC light inside the chamber when a mask is inserted. Would the energies at play per unit of surface area change? These aspects should be commented on. (Discussion)

Answer: Dear reviewer, thank you for your comment. The discussion (Pg 11-12) section has been modified according to the suggestion.

Question 21: The research is solely based on testing one virus. The resistance to other pathogens could be at least estimated based on the literature and the presented results. The hypothesis of the technique used is that it may be effective with other microbial species as well. (Discussion)

Answer: Dear reviewer, thank you for your comment. The discussion (Pg 11-12) section has been modified according to the suggestion.

Question 22: With regard to the limitations of the work, more data would have given the study greater significance and, in addition, repeating the tests on several samples for each exposure time would have avoided the ambiguity of the data found in the UVGI-only decontamination test. (Discussion)

Answer: Dear reviewer, thank you for your comment. Since this comment is similar to Question 16 from Reviewer 2, the answer is consolidated together and discussed more in detail. Kindly please review the comment in the said section.

Question 23: It should be noted: UVGI reaches 3 logs (minimum required pathogen reduction to completely disinfect a contaminated) in about a quarter of the time compared to UVGI + MGS. (Conclusion)

Answer: Dear reviewer, thank you for your comment. The conclusion (Pg 12) section has been modified according to the suggestion.

Question 24: Cited References are relevant and sufficient to place the work in context. (Have a look at the suggested work in the Introduction section.) (References)

Answer: Dear reviewer, thank you for your comment. The new journal references are added accordingly.

Question 25: The tables and graph are clear. Graph needs better quality (Tables and Figures)

Answer: Dear reviewer, thank you for your comment. A new graph has been produced using the PACE tool.

Question 26: For the sake of clarity, the legend should be reproduced below the relevant figure. (Tables and Figures)

Answer: Dear reviewer, thank you for your comment. Figure 1 and 2 has been reproduced using legends attached to the figure as per suggestion. Figures have been produced using the PACE tool.

---

## [Editor Report · Decision Letter 1]

21 Dec 2023

N95 Respirator Hybrid Decontamination Method Using Ultraviolet Germicidal Irradiation (UVGI) Coupled with Microwave-Generated Steam (MGS)

PONE-D-23-26467R1

Dear Dr. Gopalan,

We’re pleased to inform you that your manuscript has been judged scientifically suitable for publication and will be formally accepted for publication once it meets all outstanding technical requirements.

Kind regards,

Vittorio Sambri, M.D., Ph.D.

Academic Editor

PLOS ONE
---

## [Editor Report · Acceptance letter]

29 Jan 2024

PONE-D-23-26467R1 

PLOS ONE

Dear Dr. Muhamad, 

I'm pleased to inform you that your manuscript has been deemed suitable for publication in PLOS ONE. Congratulations! Your manuscript is now being handed over to our production team.

Kind regards, 

on behalf of

Professor Vittorio Sambri 

Academic Editor

PLOS ONE